# Fast $\epsilon$-free Inference of Simulation Models with Bayesian Conditional Density Estimation

**George Papamakarios**
School of Informatics
University of Edinburgh
g.papamakarios@ed.ac.uk

**Iain Murray**
School of Informatics
University of Edinburgh
i.murray@ed.ac.uk

## Abstract

Many statistical models can be simulated forwards but have intractable likelihoods. Approximate Bayesian Computation (ABC) methods are used to infer properties of these models from data. Traditionally these methods approximate the posterior over parameters by conditioning on data being inside an $\epsilon$-ball around the observed data, which is only correct in the limit $\epsilon \to 0$. Monte Carlo methods can then draw samples from the approximate posterior to approximate predictions or error bars on parameters. These algorithms critically slow down as $\epsilon \to 0$, and in practice draw samples from a broader distribution than the posterior. We propose a new approach to likelihood-free inference based on Bayesian conditional density estimation. Preliminary inferences based on limited simulation data are used to guide later simulations. In some cases, learning an accurate parametric representation of the entire true posterior distribution requires fewer model simulations than Monte Carlo ABC methods need to produce a single sample from an approximate posterior.

## 1 Introduction

A simulator-based model is a data-generating process described by a computer program, usually with some free parameters we need to learn from data. Simulator-based modelling lends itself naturally to scientific domains such as evolutionary biology [1], ecology [24], disease epidemics [10], economics [8] and cosmology [23], where observations are best understood as products of underlying physical processes. Inference in these models amounts to discovering plausible parameter settings that could have generated our observed data. The application domains mentioned can require properly calibrated distributions that express uncertainty over plausible parameters, rather than just point estimates, in order to reach scientific conclusions or make decisions.

As an analytical expression for the likelihood of parameters given observations is typically not available for simulator-based models, conventional likelihood-based Bayesian inference is not applicable. An alternative family of algorithms for likelihood-free inference has been developed, referred to as Approximate Bayesian Computation (ABC). These algorithms simulate the model repeatedly and only accept parameter settings which generate synthetic data similar to the observed data, typically gathered in a real-world experiment.

Rejection ABC [21], the most basic ABC algorithm, simulates the model for each setting of proposed parameters, and rejects parameters if the generated data is not within a certain distance from the observations. The accepted parameters form a set of independent samples from an approximate posterior. Markov Chain Monte Carlo ABC (MCMC-ABC) [13] is an improvement over rejection ABC which, instead of independently proposing parameters, explores the parameter space by perturbing the most recently accepted parameters. Sequential Monte Carlo ABC (SMC-ABC) [2, 5] uses importance sampling to simulate a sequence of slowly-changing distributions, the last of which is an approximation to the parameter posterior.

Conventional ABC algorithms such as the above suffer from three drawbacks. First, they only represent the parameter posterior as a set of (possibly weighted or correlated) samples. A sample-based representation easily gives estimates and error bars of individual parameters, and model predictions. However these computations are noisy, and it is not obvious how to perform some other computations using samples, such as combining posteriors from two separate analyses. Second, the parameter samples do not come from the correct Bayesian posterior, but from an approximation based on assuming a pseudo-observation that the data is within an $\epsilon$-ball centred on the data actually observed. Third, as the $\epsilon$-tolerance is reduced, it can become impractical to simulate the model enough times to match the observed data even once. When simulations are expensive to perform, good quality inference becomes impractical.

We propose a parametric approach to likelihood-free inference, which unlike conventional ABC does not suffer from the above three issues. Instead of returning samples from an $\epsilon$-approximation to the posterior, our approach learns a parametric approximation to the exact posterior, which can be made as accurate as required. Preliminary fits to the posterior are used to guide future simulations, which can reduce the number of simulations required to learn an accurate approximation by orders of magnitude. Our approach uses conditional density estimation with Bayesian neural networks, and draws upon advances in parametric density estimation, stochastic variational inference, and recognition networks, as discussed in the related work section.

## 2 Bayesian conditional density estimation for likelihood-free inference

### 2.1 Simulator-based models and ABC

Let $\boldsymbol{\theta}$ be a vector of parameters controlling a simulator-based model, and let $\mathbf{x}$ be a data vector generated by the model. The model may be provided as a probabilistic program that can be easily simulated, and implicitly defines a likelihood $p(\mathbf{x} \,|\, \boldsymbol{\theta})$, which we assume we cannot evaluate. Let $p(\boldsymbol{\theta})$ encode our prior beliefs about the parameters. Given an observation $\mathbf{x}_o$, we are interested in the parameter posterior $p(\boldsymbol{\theta} \,|\, \mathbf{x} = \mathbf{x}_o) \propto p(\mathbf{x} = \mathbf{x}_o \,|\, \boldsymbol{\theta})\, p(\boldsymbol{\theta})$.

As the likelihood $p(\mathbf{x} = \mathbf{x}_o \,|\, \boldsymbol{\theta})$ is unavailable, conventional Bayesian inference cannot be carried out. The principle behind ABC is to approximate $p(\mathbf{x} = \mathbf{x}_o \,|\, \boldsymbol{\theta})$ by $p(\|\mathbf{x} - \mathbf{x}_o\| < \epsilon \,|\, \boldsymbol{\theta})$ for a sufficiently small value of $\epsilon$, and then estimate the latter—e.g. by Monte Carlo—using simulations from the model. Hence, ABC approximates the posterior by $p(\boldsymbol{\theta} \,|\, \|\mathbf{x} - \mathbf{x}_o\| < \epsilon)$, which is typically broader and more uncertain. ABC can trade off computation for accuracy by decreasing $\epsilon$, which improves the approximation to the posterior but requires more simulations from the model. However, the approximation becomes exact only when $\epsilon \to 0$, in which case simulations never match the observations, $p(\|\mathbf{x} - \mathbf{x}_o\| < \epsilon \,|\, \boldsymbol{\theta}) \to 0$, and existing methods break down. In this paper, we refer to $p(\boldsymbol{\theta} \,|\, \mathbf{x} = \mathbf{x}_o)$ as the *exact posterior*, as it corresponds to setting $\epsilon = 0$ in $p(\boldsymbol{\theta} \,|\, \|\mathbf{x} - \mathbf{x}_o\| < \epsilon)$.

In most practical applications of ABC, $\mathbf{x}$ is taken to be a fixed-length vector of summary statistics that is calculated from data generated by the simulator, rather than the raw data itself. Extracting statistics is often necessary in practice, to reduce the dimensionality of the data and maintain $p(\|\mathbf{x} - \mathbf{x}_o\| < \epsilon \,|\, \boldsymbol{\theta})$ to practically acceptable levels. For the purposes of this paper, we will make no distinction between raw data and summary statistics, and we will regard the calculation of summary statistics as part of the data generating process.

### 2.2 Learning the posterior

Rather than using simulations from the model in order to estimate an approximate likelihood, $p(\|\mathbf{x} - \mathbf{x}_o\| < \epsilon \,|\, \boldsymbol{\theta})$, we will use the simulations to directly estimate $p(\boldsymbol{\theta} \,|\, \mathbf{x} = \mathbf{x}_o)$. We will run simulations for parameters drawn from a distribution, $\tilde{p}(\boldsymbol{\theta})$, which we shall refer to as the *proposal prior*. The proposition below indicates how we can then form a consistent estimate of the exact posterior, using a flexible family of conditional densities, $q_\phi(\boldsymbol{\theta} \,|\, \mathbf{x})$, parameterized by a vector $\phi$.

**Proposition 1.** *We assume that each of a set of $N$ pairs $(\boldsymbol{\theta}_n, \mathbf{x}_n)$ was independently generated by*

$$\boldsymbol{\theta}_n \sim \tilde{p}(\boldsymbol{\theta}) \quad and \quad \mathbf{x}_n \sim p(\mathbf{x} \,|\, \boldsymbol{\theta}_n). \tag{1}$$

*In the limit $N \to \infty$, the probability of the parameter vectors $\prod_n q_\phi(\boldsymbol{\theta}_n \,|\, \mathbf{x}_n)$ is maximized w.r.t. $\phi$ if and only if*

$$q_\phi(\boldsymbol{\theta} \,|\, \mathbf{x}) \propto \frac{\tilde{p}(\boldsymbol{\theta})}{p(\boldsymbol{\theta})}\, p(\boldsymbol{\theta} \,|\, \mathbf{x}), \tag{2}$$

*provided a setting of $\phi$ that makes $q_\phi(\boldsymbol{\theta} \mid \mathbf{x})$ proportional to $\frac{\tilde{p}(\boldsymbol{\theta})}{p(\boldsymbol{\theta})} p(\boldsymbol{\theta} \mid \mathbf{x})$ exists.*

*Intuition*: if we simulated enough parameters from the prior, the density estimator $q_\phi$ would learn a conditional of the joint prior model over parameters and data, which is the posterior $p(\boldsymbol{\theta} \mid \mathbf{x})$. If we simulate parameters drawn from another distribution, we need to "importance reweight" the result. A more detailed proof can be found in Section A of the supplementary material.

The proposition above suggests the following procedure for learning the posterior: (a) propose a set of parameter vectors $\{\boldsymbol{\theta}_n\}$ from the proposal prior; (b) for each $\boldsymbol{\theta}_n$ run the simulator to obtain a corresponding data vector $\mathbf{x}_n$; (c) train $q_\phi$ with maximum likelihood on $\{\boldsymbol{\theta}_n, \mathbf{x}_n\}$; and (d) estimate the posterior by

$$\hat{p}(\boldsymbol{\theta} \mid \mathbf{x} = \mathbf{x}_o) \propto \frac{p(\boldsymbol{\theta})}{\tilde{p}(\boldsymbol{\theta})} \, q_\phi(\boldsymbol{\theta} \mid \mathbf{x}_o). \tag{3}$$

This procedure is summarized in Algorithm 2.

## 2.3   Choice of conditional density estimator and proposal prior

In choosing the types of density estimator $q_\phi(\boldsymbol{\theta} \mid \mathbf{x})$ and proposal prior $\tilde{p}(\boldsymbol{\theta})$, we need to meet the following criteria: (a) $q_\phi$ should be flexible enough to represent the posterior but easy to train with maximum likelihood; (b) $\tilde{p}(\boldsymbol{\theta})$ should be easy to evaluate and sample from; and (c) the right-hand side expression in Equation (3) should be easily evaluated and normalized.

We draw upon work on conditional neural density estimation and take $q_\phi$ to be a Mixture Density Network (MDN) [3] with fully parameterized covariance matrices. That is, $q_\phi$ takes the form of a mixture of $K$ Gaussian components $q_\phi(\boldsymbol{\theta} \mid \mathbf{x}) = \sum_k \alpha_k \, \mathcal{N}(\boldsymbol{\theta} \mid \mathbf{m}_k, \mathbf{S}_k)$, whose mixing coefficients $\{\alpha_k\}$, means $\{\mathbf{m}_k\}$ and covariance matrices $\{\mathbf{S}_k\}$ are computed by a feedforward neural network parameterized by $\phi$, taking $\mathbf{x}$ as input. Such an architecture is capable of representing any conditional distribution arbitrarily accurately—provided the number of components $K$ and number of hidden units in the neural network are sufficiently large—while remaining trainable by backpropagation. The parameterization of the MDN is detailed in Section B of the supplementary material.

We take the proposal prior to be a single Gaussian $\tilde{p}(\boldsymbol{\theta}) = \mathcal{N}(\boldsymbol{\theta} \mid \mathbf{m}_0, \mathbf{S}_0)$, with mean $\mathbf{m}_0$ and full covariance matrix $\mathbf{S}_0$. Assuming the prior $p(\boldsymbol{\theta})$ is a simple distribution (uniform or Gaussian, as is typically the case in practice), then this choice allows us to calculate $\hat{p}(\boldsymbol{\theta} \mid \mathbf{x} = \mathbf{x}_o)$ in Equation (3) analytically. That is, $\hat{p}(\boldsymbol{\theta} \mid \mathbf{x} = \mathbf{x}_o)$ will be a mixture of $K$ Gaussians, whose parameters will be a function of $\{\alpha_k, \mathbf{m}_k, \mathbf{S}_k\}$ evaluated at $\mathbf{x}_o$ (as detailed in Section C of the supplementary material).

## 2.4   Learning the proposal prior

Simple rejection ABC is inefficient because the posterior $p(\boldsymbol{\theta} \mid \mathbf{x} = \mathbf{x}_o)$ is typically much narrower than the prior $p(\boldsymbol{\theta})$. A parameter vector $\boldsymbol{\theta}$ sampled from $p(\boldsymbol{\theta})$ will rarely be plausible under $p(\boldsymbol{\theta} \mid \mathbf{x} = \mathbf{x}_o)$ and will most likely be rejected. Practical ABC algorithms attempt to reduce the number of rejections by modifying the way they propose parameters; for instance, MCMC-ABC and SMC-ABC propose new parameters by perturbing parameters they already consider plausible, in the hope that nearby parameters remain plausible.

In our framework, the key to efficient use of simulations lies in the choice of proposal prior. If we take $\tilde{p}(\boldsymbol{\theta})$ to be the actual prior, then $q_\phi(\boldsymbol{\theta} \mid \mathbf{x})$ will learn the posterior for all $\mathbf{x}$, as can be seen from Equation (2). Such a strategy however is grossly inefficient if we are only interested in the posterior for $\mathbf{x} = \mathbf{x}_o$. Conversely, if $\tilde{p}(\boldsymbol{\theta})$ closely matches $p(\boldsymbol{\theta} \mid \mathbf{x} = \mathbf{x}_o)$, then most simulations will produce samples that are highly informative in learning $q_\phi(\boldsymbol{\theta} \mid \mathbf{x})$ for $\mathbf{x} = \mathbf{x}_o$. In other words, if we already knew the true posterior, we could use it to construct an efficient proposal prior for learning it.

We exploit this idea to set up a fixed-point system. Our strategy becomes to learn an efficient proposal prior that closely approximates the posterior as follows: (a) initially take $\tilde{p}(\boldsymbol{\theta})$ to be the prior $p(\boldsymbol{\theta})$; (b) propose $N$ samples $\{\boldsymbol{\theta}_n\}$ from $\tilde{p}(\boldsymbol{\theta})$ and corresponding samples $\{\mathbf{x}_n\}$ from the simulator, and train $q_\phi(\boldsymbol{\theta} \mid \mathbf{x})$ on them; (c) approximate the posterior using Equation (3) and set $\tilde{p}(\boldsymbol{\theta})$ to it; (d) repeat until $\tilde{p}(\boldsymbol{\theta})$ has converged. This procedure is summarized in Algorithm 1.

In the procedure above, as long as $q_\phi(\boldsymbol{\theta} \mid \mathbf{x})$ has only one Gaussian component ($K = 1$) then $\tilde{p}(\boldsymbol{\theta})$ remains a single Gaussian throughout. Moreover, in each iteration we initialize $q_\phi$ with the density

| **Algorithm 1:** Training of proposal prior | **Algorithm 2:** Training of posterior |
|---|---|
| initialize $q_\phi(\boldsymbol{\theta} \mid \mathbf{x})$ with one component<br>$\tilde{p}(\boldsymbol{\theta}) \leftarrow p(\boldsymbol{\theta})$<br>**repeat**<br>    **for** $n = 1..N$ **do**<br>        sample $\boldsymbol{\theta}_n \sim \tilde{p}(\boldsymbol{\theta})$<br>        sample $\mathbf{x}_n \sim p(\mathbf{x} \mid \boldsymbol{\theta}_n)$<br>    **end**<br>    retrain $q_\phi(\boldsymbol{\theta} \mid \mathbf{x})$ on $\{\boldsymbol{\theta}_n, \mathbf{x}_n\}$<br>    $\tilde{p}(\boldsymbol{\theta}) \leftarrow \frac{p(\boldsymbol{\theta})}{\tilde{p}(\boldsymbol{\theta})} q_\phi(\boldsymbol{\theta} \mid \mathbf{x}_o)$<br>**until** $\tilde{p}(\boldsymbol{\theta})$ *has converged*; | initialize $q_\phi(\boldsymbol{\theta} \mid \mathbf{x})$ with K components<br>`// if` $q_\phi$ `available by Algorithm 1`<br>`// initialize by replicating its`<br>`// one component` $K$ `times`<br>**for** $n = 1..N$ **do**<br>    sample $\boldsymbol{\theta}_n \sim \tilde{p}(\boldsymbol{\theta})$<br>    sample $\mathbf{x}_n \sim p(\mathbf{x} \mid \boldsymbol{\theta}_n)$<br>**end**<br>train $q_\phi(\boldsymbol{\theta} \mid \mathbf{x})$ on $\{\boldsymbol{\theta}_n, \mathbf{x}_n\}$<br>$\hat{p}(\boldsymbol{\theta} \mid \mathbf{x} = \mathbf{x}_o) \leftarrow \frac{p(\boldsymbol{\theta})}{\tilde{p}(\boldsymbol{\theta})} q_\phi(\boldsymbol{\theta} \mid \mathbf{x}_o)$ |

estimator learnt in the iteration before, thus we keep training $q_\phi$ throughout. This initialization allows us to use a small sample size $N$ in each iteration, thus making efficient use of simulations.

As we shall demonstrate in Section 3, the procedure above learns Gaussian approximations to the true posterior fast: in our experiments typically 4–6 iterations of 200–500 samples each were sufficient. This Gaussian approximation can be used as a rough but cheap approximation to the true posterior, or it can serve as a good proposal prior in Algorithm 2 for efficiently fine-tuning a non-Gaussian multi-component posterior. If the second strategy is adopted, then we can reuse the single-component neural density estimator learnt in Algorithm 1 to initialize $q_\phi$ in Algorithm 2. The weights in the final layer of the MDN are replicated $K$ times, with small random perturbations to break symmetry.

## 2.5 Use of Bayesian neural density estimators

To make Algorithm 1 as efficient as possible, the number of simulations per iteration $N$ should be kept small, while at the same time it should provide a sufficient training signal for $q_\phi$. With a conventional MDN, if $N$ is made too small, there is a danger of overfitting, especially in early iterations, leading to over-confident proposal priors and an unstable procedure. Early stopping could be used to avoid overfitting; however a significant fraction of the $N$ samples would have to be used as a validation set, leading to inefficient use of simulations.

As a better alternative, we developed a Bayesian version of the MDN using Stochastic Variational Inference (SVI) for neural networks [12]. We shall refer to this Bayesian version of the MDN as MDN-SVI. An MDN-SVI has two sets of adjustable parameters of the same size, the means $\phi_m$ and the log variances $\phi_s$. The means correspond to the parameters $\phi$ of a conventional MDN. During training, Gaussian noise of variance $\exp \phi_s$ is added to the means independently for each training example $(\boldsymbol{\theta}_n, \mathbf{x}_n)$. The Bayesian interpretation of this procedure is that it optimizes a variational Gaussian posterior with a diagonal covariance matrix over parameters $\phi$. At prediction time, the noise is switched off and the MDN-SVI behaves like a conventional MDN with $\phi = \phi_m$. Section D of the supplementary material details the implementation and training of MDN-SVI. We found that using an MDN-SVI instead of an MDN improves the robustness and efficiency of Algorithm 1 because (a) MDN-SVI is resistant to overfitting, allowing us to use a smaller number of simulations $N$; (b) no validation set is needed, so all samples can be used for training; and (c) since overfitting is not an issue, no careful tuning of training time is necessary.

## 3 Experiments

We showcase three versions of our approach: (a) learning the posterior with Algorithm 2 where $q_\phi$ is a conventional MDN and the proposal prior $\tilde{p}(\boldsymbol{\theta})$ is taken to be the actual prior $p(\boldsymbol{\theta})$, which we refer to as *MDN with prior*; (b) training a proposal prior with Algorithm 1 where $q_\phi$ is an MDN-SVI, which we refer to as *proposal prior*; and (c) learning the posterior with Algorithm 2 where $q_\phi$ is an MDN-SVI and the proposal prior $\tilde{p}(\boldsymbol{\theta})$ is taken to be the one learnt in (b), which we refer to as *MDN with proposal*. All MDNs were trained using Adam [11] with its default parameters.

We compare to three ABC baselines: (a) rejection ABC [21], where parameters are proposed from the prior and are accepted if $\|\mathbf{x} - \mathbf{x}_o\| < \epsilon$; (b) MCMC-ABC [13] with a spherical Gaussian proposal, whose variance we manually tuned separately in each case for best performance; and (c) SMC-

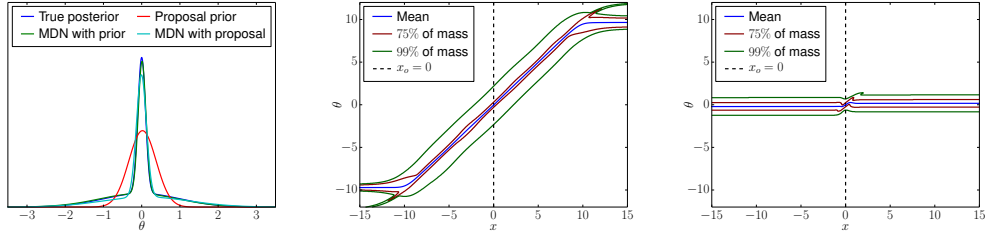

Figure 1: Results on mixture of two Gaussians. **Left**: approximate posteriors learnt by each strategy for $x_o = 0$. **Middle**: full conditional density $q_\phi(\theta|x)$ leant by the MDN trained with prior. **Right**: full conditional density $q_\phi(\theta|x)$ learnt by the MDN-SVI trained with proposal prior. Vertical dashed lines show the location of the observation $x_o = 0$.

ABC [2], where the sequence of $\epsilon$'s was exponentially decayed, with a decay rate manually tuned separately in each case for best performance. MCMC-ABC was given the unrealistic advantage of being initialized with a sample from rejection ABC, removing the need for an otherwise necessary burn-in period. Code for reproducing the experiments is provided in the supplementary material and at `https://github.com/gpapamak/epsilon_free_inference`.

## 3.1 Mixture of two Gaussians

The first experiment is a toy problem where the goal is to infer the common mean $\theta$ of a mixture of two 1D Gaussians, given a single datapoint $x_o$. The setup is

$$p(\theta) = \mathcal{U}(\theta \,|\, \theta_\alpha, \theta_\beta) \quad \text{and} \quad p(x \,|\, \theta) = \alpha \mathcal{N}\big(x \,|\, \theta, \sigma_1^2\big) + (1 - \alpha) \mathcal{N}\big(x \,|\, \theta, \sigma_2^2\big), \quad (4)$$

where $\theta_\alpha = -10$, $\theta_\beta = 10$, $\alpha = 0.5$, $\sigma_1 = 1$, $\sigma_2 = 0.1$ and $x_o = 0$. The posterior can be calculated analytically, and is proportional to an equal mixture of two Gaussians centred at $x_o$ with variances $\sigma_1^2$ and $\sigma_2^2$, restricted to $[\theta_\alpha, \theta_\beta]$. This problem is often used in the SMC-ABC literature to illustrate the difficulty of MCMC-ABC in representing long tails. Here we use it to demonstrate the correctness of our approach and its ability to accurately represent non-Gaussian long-tailed posteriors.

Figure 1 shows the results of neural density estimation using each strategy. All MDNs have one hidden layer with 20 tanh units and 2 Gaussian components, except for the proposal prior MDN which has a single component. Both MDN with prior and MDN with proposal learn good parametric approximations to the true posterior, and the proposal prior is a good Gaussian approximation to it. We used 10K simulations to train the MDN with prior, whereas the prior proposal took 4 iterations of 200 simulations each to train, and the MDN with proposal took 1000 simulations on top of the previous 800. The MDN with prior learns the posterior distributions for a large range of possible observations $x$ (middle plot of Figure 1), whereas the MDN with proposal gives accurate posterior probabilities only near the value actually observed (right plot of Figure 1).

## 3.2 Bayesian linear regression

In Bayesian linear regression, the goal is to infer the parameters $\boldsymbol{\theta}$ of a linear map from noisy observations of outputs at known inputs. The setup is

$$p(\boldsymbol{\theta}) = \mathcal{N}(\boldsymbol{\theta} \,|\, \mathbf{m}, \mathbf{S}) \quad \text{and} \quad p(\mathbf{x} \,|\, \boldsymbol{\theta}) = \prod_i \mathcal{N}\big(x_i \,|\, \boldsymbol{\theta}^T \mathbf{u}_i, \sigma^2\big), \quad (5)$$

where we took $\mathbf{m} = \mathbf{0}$, $\mathbf{S} = \mathbf{I}$, $\sigma = 0.1$, randomly generated inputs $\{\mathbf{u}_i\}$ from a standard Gaussian, and randomly generated observations $\mathbf{x}_o$ from the model. In our setup, $\boldsymbol{\theta}$ and $\mathbf{x}$ have 6 and 10 dimensions respectively. The posterior is analytically tractable, and is a single Gaussian.

All MDNs have one hidden layer of 50 tanh units and one Gaussian component. ABC methods were run for a sequence of decreasing $\epsilon$'s, up to their failing points. To measure the approximation quality to the posterior, we analytically calculated the KL divergence from the true posterior to the learnt posterior (which for ABC was taken to be a Gaussian fit to the set of returned posterior samples). The left of Figure 2 shows the approximation quality vs $\epsilon$; MDN methods are shown as horizontal

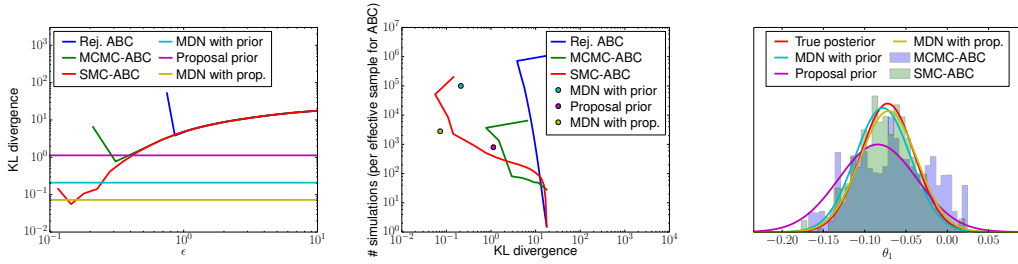

Figure 2: Results on Bayesian linear regression. **Left**: KL divergence from true posterior to approximation vs $\epsilon$; lower is better. **Middle**: number of simulations vs KL divergence; lower left is better. Note that number of simulations is total for MDNs, and per effective sample for ABC. **Right**: Posterior marginals for $\theta_1$ as computed by each method. ABC posteriors (represented as histograms) correspond to the setting of $\epsilon$ that minimizes the KL in the left plot.

lines. As $\epsilon$ is decreased, ABC methods sample from an increasingly better approximation to the true posterior, however they eventually reach their failing point, or take prohibitively long. The best approximations are achieved by MDN with proposal and a very long run of SMC-ABC.

The middle of Figure 2 shows the increase in number of simulations needed to improve approximation quality (as $\epsilon$ decreases). We quote the *total* number of simulations for MDN training, and the number of simulations *per effective sample* for ABC. Section E of the supplementary material describes how the number of effective samples is calculated. The number of simulations per effective sample should be multiplied by the number of effective samples needed in practice. Moreover, SMC-ABC will not work well with only one particle, so many times the quoted cost will always be needed. Here, MDNs make more efficient use of simulations than Monte Carlo ABC methods. Sequentially fitting a prior proposal was more than ten times cheaper than training with prior samples, and more accurate.

### 3.3 Lotka–Volterra predator-prey population model

The Lotka–Volterra model is a stochastic Markov jump process that describes the continuous time evolution of a population of predators interacting with a population of prey. There are four possible reactions: (a) a predator being born, (b) a predator dying, (c) a prey being born, and (d) a prey being eaten by a predator. Positive parameters $\boldsymbol{\theta} = (\theta_1, \theta_2, \theta_3, \theta_4)$ control the rate of each reaction. Given a set of statistics $\mathbf{x}_o$ calculated from an observed population time series, the objective is to infer $\boldsymbol{\theta}$. We used a flat prior over $\log \boldsymbol{\theta}$, and calculated a set of 9 statistics $\mathbf{x}$. The full setup is detailed in Section F of the supplementary material. The Lotka–Volterra model is commonly used in the ABC literature as a realistic model which can be simulated, but whose likelihood is intractable. One of the properties of Lotka–Volterra is that typical nature-like observations only occur for very specific parameter settings, resulting in narrow, Gaussian-like posteriors that are hard to recover.

The MDN trained with prior has two hidden layers of 50 tanh units each, whereas the MDN-SVI used to train the proposal prior and the MDN-SVI trained with proposal have one hidden layer of 50 tanh units. All three have one Gaussian component. We found that using more than one components made no difference to the results; in all cases the MDNs chose to use only one component and switch the rest off, which is consistent with our observation about the near-Gaussianity of the posterior.

We measure how well each method retrieves the true parameter values that were used to generate $\mathbf{x}_o$ by calculating their log probability under each learnt posterior; for ABC a Gaussian fit to the posterior samples was used. The left panel of Figure 3 shows how this log probability varies with $\epsilon$, demonstrating the superiority of MDN methods over ABC. In the middle panel we can see that MDN training with proposal makes efficient use of simulations compared to training with prior and ABC; note that for ABC the number of simulations is only for *one effective sample*. In the right panel, we can see that the estimates returned by MDN methods are more confident around the true parameters compared to ABC, because the MDNs learn the exact posterior rather than an inflated version of it like ABC does (plots for the other three parameters look similar).

We found that when training an MDN with a well-tuned proposal that focuses on the plausible region, an MDN with fewer parameters is needed compared to training with the prior. This is because the

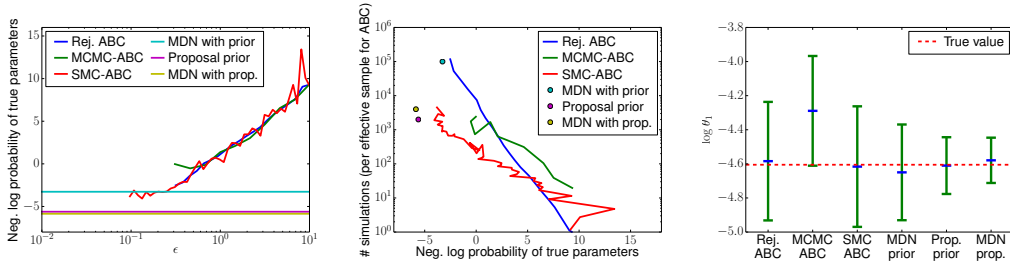

Figure 3: Results on Lotka–Volterra. **Left**: negative log probability of true parameters vs $\epsilon$; lower is better. **Middle**: number of simulations vs negative log probability; lower left is better. Note that number of simulations is total for MDNs, but per effective sample for ABC. **Right**: Estimates of $\log \theta_1$ with 2 standard deviations. ABC estimates used many more simulations with the smallest feasible $\epsilon$.

MDN trained with proposal needs to learn only the *local* relationship between $\mathbf{x}$ and $\boldsymbol{\theta}$ near $\mathbf{x}_o$, as opposed to in the entire domain of the prior. Hence, not only are savings achieved in number of simulations, but also training the MDN itself becomes more efficient.

### 3.4 M/G/1 queue model

The M/G/1 queue model describes the processing of a queue of continuously arriving jobs by a single server. In this model, the time the server takes to process each job is independently and uniformly distributed in the interval $[\theta_1, \theta_2]$. The time interval between arrival of two consecutive jobs is independently and exponentially distributed with rate $\theta_3$. The server observes only the time intervals between departure of two consecutive jobs. Given a set of equally-spaced percentiles $\mathbf{x}_o$ of inter-departure times, the task is to infer parameters $\boldsymbol{\theta} = (\theta_1, \theta_2, \theta_3)$. This model is easy to simulate but its likelihood is intractable, and it has often been used as an ABC benchmark [4, 16]. Unlike Lotka–Volterra, data $\mathbf{x}$ is weakly informative about $\boldsymbol{\theta}$, and hence the posterior over $\boldsymbol{\theta}$ tends to be broad and non-Gaussian. In our setup, we placed flat independent priors over $\theta_1$, $\theta_2 - \theta_1$ and $\theta_3$, and we took $\mathbf{x}$ to be $5$ equally spaced percentiles, as detailed in Section G of the supplementary material.

The MDN trained with prior has two hidden layers of $50$ tanh units each, whereas the MDN-SVI used to train the proposal prior and the one trained with proposal have one hidden layer of $50$ tanh units. As observed in the Lotka–Volterra demo, less capacity is required when training with proposal, as the relationship to be learned is local and hence simpler, which saves compute time and gives a more accurate final posterior. All MDNs have $8$ Gaussian components (except the MDN-SVI used to train the proposal prior, which always has one), which, after experimentation, we determined are enough for the MDNs to represent the non-Gaussian nature of the posterior.

Figure 4 reports the log probability of the true parameters under each posterior learnt—for ABC, the log probability was calculated by fitting a mixture of $8$ Gaussians to posterior samples using Expectation-Maximization—and the number of simulations needed to achieve it. As before, MDN methods are more confident compared to ABC around the true parameters, which is due to ABC computing a broader posterior than the true one. MDN methods make more efficient use of simulations, since they use all of them for training and, unlike ABC, do not throw a proportion of them away.

## 4  Related work

**Regression adjustment**. An early parametric approach to ABC is *regression adjustment*, where a parametric regressor is trained on simulation data in order to learn a mapping from $\mathbf{x}$ to $\boldsymbol{\theta}$. The learnt mapping is then used to correct for using a large $\epsilon$, by adjusting the location of posterior samples gathered by e.g. rejection ABC. Beaumont et al. [1] used linear regressors, and later Blum and François [4] used neural networks with one hidden layer that separately predicted the mean and variance of $\boldsymbol{\theta}$. Both can be viewed as rudimentary density estimators and as such they are a predecessor to our work. However, they were not flexible enough to accurately estimate the posterior, and they were only used within some other ABC method to allow for a larger $\epsilon$. In our work, we make conditional density estimation flexible enough to approximate the posterior accurately.

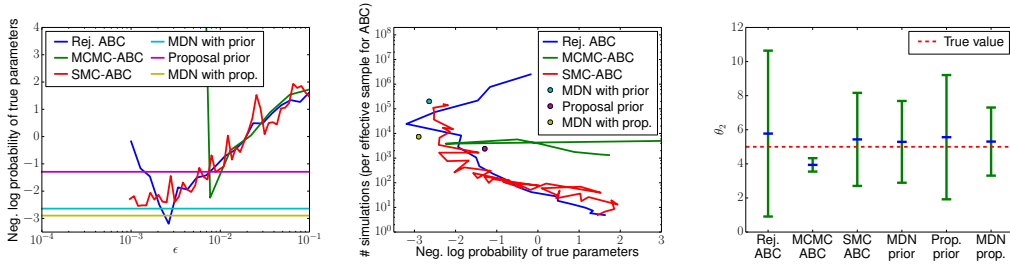

Figure 4: Results on M/G/1. **Left**: negative log probability of true parameters vs $\epsilon$; lower is better. **Middle**: number of simulations vs negative log probability; lower left is better. Note that number of simulations is total for MDNs, and per effective sample for ABC. **Right**: Estimates of $\theta_2$ with 2 standard deviations; ABC estimates correspond to the lowest setting of $\epsilon$ used.

**Synthetic likelihood**. Another parametric approach is *synthetic likelihood*, where parametric models are used to estimate the likelihood $p(\mathbf{x} \,|\, \boldsymbol{\theta})$. Wood [24] used a single Gaussian, and later Fan et al. [7] used a mixture Gaussian model. Both of them learnt a separate density model of $\mathbf{x}$ for each $\boldsymbol{\theta}$ by repeatedly simulating the model for fixed $\boldsymbol{\theta}$. More recently, Meeds and Welling [14] used a Gaussian process model to interpolate Gaussian likelihood approximations between different $\boldsymbol{\theta}$'s. Compared to learning the posterior, synthetic likelihood has the advantage of not depending on the choice of proposal prior. Its main disadvantage is the need of further approximate inference on top of it in order to obtain the posterior. In our work we directly learn the posterior, eliminating the need for further inference, and we address the problem of correcting for the proposal prior.

**Efficient Monte Carlo ABC**. Recent work on ABC has focused on reducing the simulation cost of sample-based ABC methods. Hamiltonian ABC [15] improves upon MCMC-ABC by using stochastically estimated gradients in order to explore the parameter space more efficiently. Optimization Monte Carlo ABC [16] explicitly optimizes the location of ABC samples, which greatly reduces rejection rate. Bayesian optimization ABC [10] models $p(\|\mathbf{x} - \mathbf{x}_o\| \,|\, \boldsymbol{\theta})$ as a Gaussian process and then uses Bayesian optimization to guide simulations towards the region of small distances $\|\mathbf{x} - \mathbf{x}_o\|$. In our work we show how a significant reduction in simulation cost can also be achieved with parametric methods, which target the posterior directly.

**Recognition networks**. Our use of neural density estimators for learning posteriors is reminiscent of recognition networks in machine learning. A recognition network is a neural network that is trained to invert a generative model. The Helmholtz machine [6], the variational auto-encoder [12] and stochastic backpropagation [22] are examples where a recognition network is trained jointly with the generative network it is designed to invert. Feedforward neural networks have been used to invert black-box generative models [18] and binary-valued Bayesian networks [17], and convolutional neural networks have been used to invert a physics engine [25]. Our work illustrates the potential of recognition networks in the field of likelihood-free inference, where the generative model is fixed, and inference of its parameters is the goal.

**Learning proposals**. Neural density estimators have been employed in learning proposal distributions for importance sampling [20] and Sequential Monte Carlo [9, 19]. Although not our focus here, our fit to the posterior could also be used within Monte Carlo inference methods. In this work we see how far we can get purely by fitting a series of conditional density estimators.

## 5  Conclusions

Bayesian conditional density estimation improves likelihood-free inference in three main ways: (a) it represents the posterior parametrically, as opposed to as a set of samples, allowing for probabilistic evaluations later on in the pipeline; (b) it targets the exact posterior, rather than an $\epsilon$-approximation of it; and (c) it makes efficient use of simulations by not rejecting samples, by interpolating between samples, and by gradually focusing on the plausible parameter region. Our belief is that neural density estimation is a tool with great potential in likelihood-free inference, and our hope is that this work helps in establishing its usefulness in the field.

**Acknowledgments**

We thank Amos Storkey for useful comments. George Papamakarios is supported by the Centre for Doctoral Training in Data Science, funded by EPSRC (grant EP/L016427/1) and the University of Edinburgh, and by Microsoft Research through its PhD Scholarship Programme.

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
