[Supplementary Material]

# Supplementary material for "Fast $\epsilon$-free Inference of Simulation Models with Bayesian Conditional Density Estimation"

**George Papamakarios**
School of Informatics
University of Edinburgh
g.papamakarios@ed.ac.uk

**Iain Murray**
School of Informatics
University of Edinburgh
i.murray@ed.ac.uk

## A  Proof of Proposition 1

Maximizing $\prod_n q_\phi(\boldsymbol{\theta}_n \,|\, \mathbf{x}_n)$ w.r.t. $\phi$ is equivalent to maximizing the average log probability

$$\frac{1}{N} \sum_n \log q_\phi(\boldsymbol{\theta}_n \,|\, \mathbf{x}_n). \tag{1}$$

Since $(\boldsymbol{\theta}_n, \mathbf{x}_n) \sim \tilde{p}(\boldsymbol{\theta})\, p(\mathbf{x} \,|\, \boldsymbol{\theta})$, due to the strong law of large numbers, as $N \to \infty$ the average log probability converges almost surely to the following expectation

$$\frac{1}{N} \sum_n \log q_\phi(\boldsymbol{\theta}_n \,|\, \mathbf{x}_n) \xrightarrow{\text{a.s.}} \langle \log q_\phi(\boldsymbol{\theta} \,|\, \mathbf{x}) \rangle_{\tilde{p}(\boldsymbol{\theta})\, p(\mathbf{x} \,|\, \boldsymbol{\theta})}. \tag{2}$$

Let $\tilde{p}(\mathbf{x})$ be a distribution over $\mathbf{x}$. Maximizing the above expectation w.r.t. $\phi$ is equivalent to minimizing

$$D_{\mathrm{KL}}(\tilde{p}(\boldsymbol{\theta})\, p(\mathbf{x} \,|\, \boldsymbol{\theta}) \,\|\, \tilde{p}(\mathbf{x})\, q_\phi(\boldsymbol{\theta} \,|\, \mathbf{x})) = - \langle \log q_\phi(\boldsymbol{\theta} \,|\, \mathbf{x}) \rangle_{\tilde{p}(\boldsymbol{\theta})\, p(\mathbf{x} \,|\, \boldsymbol{\theta})} + \mathrm{const}. \tag{3}$$

The above KL divergence is minimized (and becomes 0) if and only if

$$\tilde{p}(\boldsymbol{\theta})\, p(\mathbf{x} \,|\, \boldsymbol{\theta}) = \tilde{p}(\mathbf{x})\, q_\phi(\boldsymbol{\theta} \,|\, \mathbf{x}) \tag{4}$$

almost everywhere. It is easy to see that this can only happen for $\tilde{p}(\mathbf{x}) = \int \tilde{p}(\boldsymbol{\theta})\, p(\mathbf{x} \,|\, \boldsymbol{\theta})\, \mathrm{d}\boldsymbol{\theta}$, since

$$\tilde{p}(\boldsymbol{\theta})\, p(\mathbf{x} \,|\, \boldsymbol{\theta}) = \tilde{p}(\mathbf{x})\, q_\phi(\boldsymbol{\theta} \,|\, \mathbf{x}) \;\Rightarrow\; \int \tilde{p}(\boldsymbol{\theta})\, p(\mathbf{x} \,|\, \boldsymbol{\theta})\, \mathrm{d}\boldsymbol{\theta} = \tilde{p}(\mathbf{x}) \int q_\phi(\boldsymbol{\theta} \,|\, \mathbf{x})\, \mathrm{d}\boldsymbol{\theta} = \tilde{p}(\mathbf{x}). \tag{5}$$

Thus, taking $\tilde{p}(\mathbf{x})$ as above, and assuming a setting of $\phi$ that makes the KL equal to 0 exists, the KL is minimized if and only if we have almost everywhere that

$$q_\phi(\boldsymbol{\theta} \,|\, \mathbf{x}) = \frac{\tilde{p}(\boldsymbol{\theta})}{\tilde{p}(\mathbf{x})}\, p(\mathbf{x} \,|\, \boldsymbol{\theta}) = \frac{\tilde{p}(\boldsymbol{\theta})}{\tilde{p}(\mathbf{x})}\, \frac{p(\boldsymbol{\theta} \,|\, \mathbf{x})\, p(\mathbf{x})}{p(\boldsymbol{\theta})} \propto \frac{\tilde{p}(\boldsymbol{\theta})}{p(\boldsymbol{\theta})}\, p(\boldsymbol{\theta} \,|\, \mathbf{x}). \tag{6}$$

A corollary of the above is that

$$q_\phi(\boldsymbol{\theta} \,|\, \mathbf{x}) = \frac{\tilde{p}(\boldsymbol{\theta})\, p(\mathbf{x} \,|\, \boldsymbol{\theta})}{\int \tilde{p}(\boldsymbol{\theta})\, p(\mathbf{x} \,|\, \boldsymbol{\theta})\, \mathrm{d}\boldsymbol{\theta}}, \tag{7}$$

in other words, $q_\phi(\boldsymbol{\theta} \,|\, \mathbf{x})$ becomes what the posterior would be if the prior were $\tilde{p}(\boldsymbol{\theta})$.

# B Parameterization and training of Mixture Density Networks

A Mixture Density Network (MDN) [1] is a conditional density estimator $q_\phi(\boldsymbol{\theta} \,|\, \mathbf{x})$, which takes the form of a mixture of $K$ Gaussian components, as follows

$$q_\phi(\boldsymbol{\theta} \,|\, \mathbf{x}) = \sum_k \alpha_k \, \mathcal{N}(\boldsymbol{\theta} \,|\, \mathbf{m}_k, \mathbf{S}_k). \tag{8}$$

The mixing coefficients $\boldsymbol{\alpha} = (\alpha_1, \ldots, \alpha_K)$, means $\{\mathbf{m}_k\}$ and covariance matrices $\{\mathbf{S}_k\}$ are computed by a feedforward neural network $f_{\mathbf{W},\mathbf{b}}(\mathbf{x})$, which has input $\mathbf{x}$, weights $\mathbf{W}$ and biases $\mathbf{b}$. In particular, let the output of the neural network be

$$\mathbf{y} = f_{\mathbf{W},\mathbf{b}}(\mathbf{x}). \tag{9}$$

Then, the mixing coefficients are given by

$$\boldsymbol{\alpha} = \text{softmax}(\mathbf{W}_{\boldsymbol{\alpha}}\mathbf{y} + \mathbf{b}_{\boldsymbol{\alpha}}). \tag{10}$$

The softmax ensures that the mixing coefficients are strictly positive and sum to one. Similarly, the means are given by

$$\mathbf{m}_k = \mathbf{W}_{\mathbf{m}_k}\mathbf{y} + \mathbf{b}_{\mathbf{m}_k}. \tag{11}$$

As for the covariance matrices, we need to ensure that they are symmetric and positive definite. For this reason, instead of parameterizing the covariance matrices directly, we parameterize the Cholesky factorization of their inverses. That is, we rewrite

$$\mathbf{S}_k^{-1} = \mathbf{U}_k^T \mathbf{U}_k, \tag{12}$$

where $\mathbf{U}_k$ is parameterized to be an upper triangular matrix with strictly positive elements in the diagonal, as follows

$$\text{diag}(\mathbf{U}_k) = \exp\big(\mathbf{W}_{\text{diag}(\mathbf{U}_k)}\mathbf{y} + \mathbf{b}_{\text{diag}(\mathbf{U}_k)}\big) \tag{13}$$

$$\text{utri}(\mathbf{U}_k) = \mathbf{W}_{\text{utri}(\mathbf{U}_k)}\mathbf{y} + \mathbf{b}_{\text{utri}(\mathbf{U}_k)} \tag{14}$$

$$\text{ltri}(\mathbf{U}_k) = \mathbf{0}. \tag{15}$$

In the above, $\text{diag}(\cdot)$ picks out the diagonal elements, whereas $\text{utri}(\cdot)$ and $\text{ltri}(\cdot)$ pick out the elements above and below the diagonal respectively. We chose to parameterize the factorization of $\mathbf{S}_k^{-1}$ rather than that of $\mathbf{S}_k$, since it is the inverse covariance that directly appears in the calculation of $\mathcal{N}(\boldsymbol{\theta} \,|\, \mathbf{m}_k, \mathbf{S}_k)$. Apart from ensuring the symmetry and positive definiteness of $\mathbf{S}_k$, the above parameterization also allows for efficiently calculating the log determinant of $\mathbf{S}_k$ as follows

$$-\frac{1}{2}\log\det(\mathbf{S}_k) = \text{sum}\big(\mathbf{W}_{\text{diag}(\mathbf{U}_k)}\mathbf{y} + \mathbf{b}_{\text{diag}(\mathbf{U}_k)}\big). \tag{16}$$

The above parameterization of the covariance matrix was introduced by Williams [12] for learning conditional Gaussians.

Given a set of training data $\{\boldsymbol{\theta}_n, \mathbf{x}_n\}$, training the MDN with maximum likelihood amounts to maximizing the average log probability

$$\frac{1}{N}\sum_n \log q_\phi(\boldsymbol{\theta}_n \,|\, \mathbf{x}_n) \tag{17}$$

with respect to the MDN parameters

$$\phi = \big(\mathbf{W}, \mathbf{b}, \mathbf{W}_{\boldsymbol{\alpha}}, \mathbf{b}_{\boldsymbol{\alpha}}, \{\mathbf{W}_{\mathbf{m}_k}, \mathbf{b}_{\mathbf{m}_k}, \mathbf{W}_{\text{diag}(\mathbf{U}_k)}, \mathbf{b}_{\text{diag}(\mathbf{U}_k)}, \mathbf{W}_{\text{utri}(\mathbf{U}_k)}, \mathbf{b}_{\text{utri}(\mathbf{U}_k)}\}\big). \tag{18}$$

Because the reparameterization $\phi$ described above is unconstrained, any off-the-shelf gradient-based stochastic optimizer can be used. Gradients of the average log probability can be easily computed with backpropagation. In our experiments, we implemented MDNs using Theano [8], which automatically backpropagates gradients, and we maximized the average log likelihood using Adam [3], which is currently the state of the art in minibatch-based stochastic optimization.

## C    Analytical calculation of parameter posterior

According to Proposition 1, after training $q_\phi(\boldsymbol{\theta} \,|\, \mathbf{x})$, the posterior at $\mathbf{x} = \mathbf{x}_o$ is approximated by

$$\hat{p}(\boldsymbol{\theta} \,|\, \mathbf{x} = \mathbf{x}_o) \propto \frac{p(\boldsymbol{\theta})}{\tilde{p}(\boldsymbol{\theta})} \, q_\phi(\boldsymbol{\theta} \,|\, \mathbf{x}_o). \tag{19}$$

Typically, the prior $p(\boldsymbol{\theta})$ is a simple distribution like a uniform or a Gaussian. Here we will consider the uniform case, while the Gaussian case is treated analogously. Let $p(\boldsymbol{\theta})$ be uniform everywhere (improper). Then the posterior estimate becomes

$$\hat{p}(\boldsymbol{\theta} \,|\, \mathbf{x} = \mathbf{x}_o) \propto \frac{q_\phi(\boldsymbol{\theta} \,|\, \mathbf{x}_o)}{\tilde{p}(\boldsymbol{\theta})}. \tag{20}$$

In practice, we also used this estimate for uniform priors with broad but finite support. Since $q_\phi(\boldsymbol{\theta} \,|\, \mathbf{x}_o)$ is a mixture of $K$ Gaussians and $\tilde{p}(\boldsymbol{\theta})$ is a single Gaussian, that is

$$q_\phi(\boldsymbol{\theta} \,|\, \mathbf{x}) = \sum_k \alpha_k \, \mathcal{N}(\boldsymbol{\theta} \,|\, \mathbf{m}_k, \mathbf{S}_k) \quad \text{and} \quad \tilde{p}(\boldsymbol{\theta}) = \mathcal{N}(\boldsymbol{\theta} \,|\, \mathbf{m}_0, \mathbf{S}_0), \tag{21}$$

their ratio can be calculated and normalized analytically. In particular, after some algebra it can be shown that the posterior estimate $\hat{p}(\boldsymbol{\theta} \,|\, \mathbf{x} = \mathbf{x}_o)$ is also a mixture of $K$ Gaussians

$$\hat{p}(\boldsymbol{\theta} \,|\, \mathbf{x} = \mathbf{x}_o) = \sum_k \alpha'_k \, \mathcal{N}(\boldsymbol{\theta} \,|\, \mathbf{m}'_k, \mathbf{S}'_k), \tag{22}$$

whose parameters are

$$\mathbf{S}'_k = \left( \mathbf{S}_k^{-1} - \mathbf{S}_0^{-1} \right)^{-1} \tag{23}$$

$$\mathbf{m}'_k = \mathbf{S}'_k \left( \mathbf{S}_k^{-1} \mathbf{m}_k - \mathbf{S}_0^{-1} \mathbf{m}_0 \right) \tag{24}$$

$$\alpha'_k = \frac{\alpha_k \exp\left( -\frac{1}{2} c_k \right)}{\sum_{k'} \alpha_{k'} \exp\left( -\frac{1}{2} c_{k'} \right)}, \tag{25}$$

where quantities $\{c_k\}$ are given by

$$c_k = \log \det \mathbf{S}_k - \log \det \mathbf{S}_0 - \log \det \mathbf{S}'_k + \mathbf{m}_k^T \mathbf{S}_k^{-1} \mathbf{m}_k - \mathbf{m}_0^T \mathbf{S}_0^{-1} \mathbf{m}_0 - \mathbf{m}_k'^T \mathbf{S}_k'^{-1} \mathbf{m}'_k. \tag{26}$$

For the above mixture to be well defined, the covariance matrices $\{\mathbf{S}'_k\}$ must be positive definite. This will not be the case if the proposal prior $\tilde{p}(\boldsymbol{\theta})$ is narrower than some component of $q_\phi(\boldsymbol{\theta} \,|\, \mathbf{x}_o)$ along some dimension. However, in both Algorithms 1 and 2, $q_\phi(\boldsymbol{\theta} \,|\, \mathbf{x}_o)$ is trained on parameters sampled from $\tilde{p}(\boldsymbol{\theta})$, hence, if trained properly, it tends to be narrower than $\tilde{p}(\boldsymbol{\theta})$. Our experience with Algorithms 1 and 2 is that $\{\mathbf{S}'_k\}$ not being positive definite rarely happens, whereas it happening is an indication that the algorithm's parameters have not been set up properly.

## D    Stochastic Variational Inference for Mixture Density Networks

In this section we describe our adaptation of Stochastic Variational Inference (SVI) for neural networks [4], in order to develop a Bayesian version of MDN. The first step is to express beliefs about the MDN parameters $\phi$ as independent Gaussian random variables with means $\phi_m$ and log variances $\phi_s$. Under this interpretation we can rewrite the parameters as

$$\phi = \phi_m + \exp\left( \frac{1}{2} \phi_s \right) \odot \mathbf{u}, \tag{27}$$

where the symbol $\odot$ denotes elementwise multiplication and $\mathbf{u}$ is an unknown vector drawn from a standard normal,

$$\mathbf{u} \sim \mathcal{N}(\mathbf{u} \,|\, \mathbf{0}, \mathbf{I}). \tag{28}$$

The above parameterization induces the following variational distribution over $\phi$

$$q(\phi) = \mathcal{N}(\phi \,|\, \phi_m, \text{diag}(\exp \phi_s)), \tag{29}$$

where $\mathrm{diag}(\exp\boldsymbol{\phi}_s)$ denotes a diagonal covariance matrix whose diagonal is the vector $\exp\boldsymbol{\phi}_s$. Moreover, we place the following Bayesian prior over $\boldsymbol{\phi}$

$$p(\boldsymbol{\phi}) = \mathcal{N}\big(\boldsymbol{\phi} \,|\, \mathbf{0}, \lambda^{-1}\mathbf{I}\big). \tag{30}$$

Under this prior, before seeing any data we set the parameter means $\boldsymbol{\phi}_m$ all to zero, and the parameter log variances $\boldsymbol{\phi}_s$ all equal to $\log\lambda^{-1}$. In our experiments, we used a default value of $\lambda = 0.01$.

Given training data $\{\boldsymbol{\theta}_n, \mathbf{x}_n\}$, the objective of SVI is to optimize $\boldsymbol{\phi}_m$ and $\boldsymbol{\phi}_s$ so as to make the variational distribution $q(\boldsymbol{\phi})$ be as close as possible (in KL) to the true Bayesian posterior over $\boldsymbol{\phi}$. This objective is equivalent to maximizing a variational lower bound,

$$\frac{1}{N}\sum_n \langle\log q_{\boldsymbol{\phi}}(\boldsymbol{\theta}_n \,|\, \mathbf{x}_n)\rangle_{\mathcal{N}(\mathbf{u}\,|\,\mathbf{0},\mathbf{I})} - \frac{1}{N}D_{\mathrm{KL}}(q(\boldsymbol{\phi})\,\|\,p(\boldsymbol{\phi})), \tag{31}$$

with respect to $\boldsymbol{\phi}_m$ and $\boldsymbol{\phi}_s$. The expectations in the first term of the above can be stochastically approximated by randomly drawing $\mathbf{u}$'s from a standard normal. The KL term can be calculated analytically, which yields

$$D_{\mathrm{KL}}(q(\boldsymbol{\phi})\,\|\,p(\boldsymbol{\phi})) = \frac{\lambda}{2}\left(\|\boldsymbol{\phi}_m\|^2 + \|\exp\boldsymbol{\phi}_s\|^2\right) - \mathrm{sum}(\boldsymbol{\phi}_s) + \mathrm{const}. \tag{32}$$

The above optimization problem has been parameterized in such a way that $\boldsymbol{\phi}_m$ and $\boldsymbol{\phi}_s$ are unconstrained. Moreover, the derivatives of the variational lower bound with respect to $\boldsymbol{\phi}_m$ and $\boldsymbol{\phi}_s$ can be easily calculated with backpropagation after stochastic approximations to the expectations have been made. This allows the use of any off-the-shelf gradient-based stochastic optimizer. In our experiments, we implemented MDN-SVI in Theano [8], which automatically calculates derivatives with backpropagation, and used Adam [3] for stochastic maximization of the variational lower bound.

An important practical detail for stochastically approximating the expectation terms is the *local reparameterization trick* [5], which leverages the layered feedforward structure of the MDN. Consider any hidden or output unit in the MDN; if $a$ is its activation and $\mathbf{z}$ is the vector of its inputs, then the relationship between $a$ and $\mathbf{z}$ is always of the form

$$a = \mathbf{w}^T\mathbf{z} + b, \tag{33}$$

where $\mathbf{w}$ and $b$ are the weights and bias respectively associated with this unit. As we have seen, in the SVI framework these weights and biases are Gaussian random variables with means $\mathbf{w}_m$ and $b_m$, and log variances $\mathbf{w}_s$ and $b_s$. It is easy to see that this induces a Gaussian distribution over activation $a$, whose mean $a_m$ and variance $\exp a_s$ is given by

$$a_m = \mathbf{w}_m^T\mathbf{z} + b_m \quad \text{and} \quad \exp a_s = (\exp\mathbf{w}_s)^T(\mathbf{z}\odot\mathbf{z}) + \exp b_s, \tag{34}$$

where $\odot$ denotes elementwise multiplication. Therefore, randomly sampling $\mathbf{w}$ and $b$ in order to estimate the expectations and their gradients in the variational lower bound is equivalent to directly sampling $a$ from a Gaussian with the above mean and variance. This trick saves computation by reducing calls to the random number generator, but more importantly it reduces the variance of the stochastic approximation of the expectations (intuitively this is because less randomness is involved) and hence it makes stochastic optimization more stable and faster to converge.

## E   Effective sample size of ABC methods

Rejection ABC returns a set of *independent* samples, MCMC-ABC returns a set of *correlated* samples, and SMC-ABC returns a set of independent but *weighted* samples. To make a fair comparison between them in terms of simulation cost, we quote the number of simulations per *effective* sample, that is, the total number of simulations divided by the effective sample size of the returned set of samples.

Let $\{\boldsymbol{\theta}_n\}$ be a set of $N$ samples, not necessarily independent. The effective sample size $N_{\mathrm{eff}}$ is defined to be the number of equivalent *independent* samples that would give an estimator of equal variance. For rejection ABC $N_{\mathrm{eff}} = N$, since all returned samples are independent.

Suppose that each sample is a vector of $D$ components. For MCMC-ABC, where samples come in the form of $D$ autocorrelated sequences, we estimated the effective sample size for component $d$ as

$$N_{\mathrm{eff},d} = \frac{N}{1 + 2\sum_{l=1}^{L_d} r_{dl}}, \tag{35}$$

where $r_{dl}$ is the autocorrelation coefficient of component $d$ at lag $l$, estimated from the samples. We calculated the summation up to lag $L_d$, which corresponds to the first autocorrelation coefficient that is equal to 0. Then we took the effective sample size $N_{\text{eff}}$ to be the minimum $N_{\text{eff},d}$ across components. For a more general discussion on estimating autocorrelation time (which is equal to $N/N_{\text{eff}}$ and thus equivalent to effective sample size) see Thompson [9].

For SMC-ABC, each sample is independent but comes with a corresponding non-negative weight $w_n$. The weights have to sum to one, that is $\sum_n w_n = 1$. We estimated the effective sample size by

$$N_{\text{eff}} = \frac{1}{\sum_n w_n^2}. \tag{36}$$

It is easy to see that if $w_n = 1/N$ for all $n$ then $N_{\text{eff}} = N$, and if all weights but one are 0 then $N_{\text{eff}} = 1$. For a discussion regarding the above estimate see Nowozin [6].

## F   Setup for the Lotka–Volterra experiment

The Lotka–Volterra model [10] is a stochastic model that was developed to describe the time evolution of a population of predators interacting with a population of prey. Let $X$ be the number of predators and $Y$ be the number of prey. The model asserts that the following four reactions can take place, with corresponding rates:

(i) A predator may be born, with rate $\theta_1 XY$, increasing $X$ by one.

(ii) A predator may die, with rate $\theta_2 X$, decreasing $X$ by one.

(iii) A prey may be born, with rate $\theta_3 Y$, increasing $Y$ by one.

(iv) A prey may be eaten by a predator, with rate $\theta_4 XY$, decreasing $Y$ by one.

Given initial populations $X$ and $Y$, the above model can be simulated using Gillespie's algorithm [2], as follows:

(i) Draw the time to next reaction from an exponential distribution with rate equal to the total rate $\theta_1 XY + \theta_2 X + \theta_3 Y + \theta_4 XY$.

(ii) Select a reaction at random, with probability proportional to its rate.

(iii) Simulate the reaction, and go to step (i).

In our experiments, each simulation started with initial populations $X = 50$ and $Y = 100$, and took place for a total of 30 time units. We recorded the values of $X$ and $Y$ after every 0.2 time units, resulting in two time series of 151 values each.

Data $\mathbf{x}$ was taken to be the following set of 9 statistics calculated from the time series:

(i) The mean of each time series.

(ii) The log variance of each time series.

(iii) The autocorrelation coefficient of each time series at lag 1 and lag 2.

(iv) The cross-correlation coefficient between the two time series.

Since the above statistics have potentially very different scales, we normalized them on the basis of a pilot run. That is, we performed a pilot run of 1000 simulations, calculated and stored the mean and standard deviation of each statistic across pilot simulations, and used them in all subsequent simulations to normalize each statistic by subtracting the pilot mean and dividing by the pilot standard deviation. This choice of statistics and normalization process was taken from Wilkinson [11].

From our experience with the model we observed that typical evolutions of the predator/prey populations for randomly selected parameters $\boldsymbol{\theta} = (\theta_1, \theta_2, \theta_3, \theta_4)$ include (a) the predators quickly eating all the prey and then slowly decaying exponentially, or (b) the predators quickly dying out and then the prey growing exponentially. However, for certain carefully tuned values of $\boldsymbol{\theta}$, the two populations exhibit an oscillatory behaviour, typical of natural ecological systems. In order to generate observations $\mathbf{x}_o$ for our experimental setup, we set the parameters to

$$\theta_1 = 0.01, \quad \theta_2 = 0.5, \quad \theta_3 = 1, \quad \theta_4 = 0.01 \tag{37}$$

Figure 1: Typical oscillatory behaviour of predator/prey populations corresponding to four different simulations of the Lotka–Volterra model with parameter values $\theta_1 = 0.01$, $\theta_2 = 0.5$, $\theta_3 = 1$, and $\theta_4 = 0.01$.

and simulated the model to generate $\mathbf{x}_o$. We carefully chose parameter values that give rise to oscillatory behaviour (see Figure 1 for typical examples of population evolution corresponding to the above parameters). Since only a small subset of parameters give rise to such oscillatory behaviour, the posterior $p(\boldsymbol{\theta} \,|\, \mathbf{x} = \mathbf{x}_o)$ is expected to be tightly peaked around the true parameter values. We tested our algorithms by evaluating how well (in terms of assigned log probability) each algorithm retrieves the true parameters.

Finally, we took the prior over $\boldsymbol{\theta}$ to be uniform in the log domain. That is, the prior was taken to be

$$p(\log \boldsymbol{\theta}) \propto \prod_{i=1}^{4} \mathcal{U}(\log \theta_i \,|\, \log \theta_\alpha, \log \theta_\beta), \tag{38}$$

where $\log \theta_\alpha = -5$ and $\log \theta_\beta = 2$, which of course includes the true parameters. All our inferences where done in the log domain.

# G  Setup for the M/G/1 experiment

The M/G/1 queue model [7] is a statistical model that describes how a single server processes a queue formed by a set of continuously arriving jobs. Let $I$ be the total number of jobs to be processed, $s_i$ be the time the server takes to process job $i$, $v_i$ be the time that job $i$ entered the queue, and $d_i$ be the time that job $i$ left the queue (i.e. the time when the server finished processing it). The M/G/1 queue model asserts that for each job $i$ we have

$$s_i \sim \mathcal{U}(\theta_1, \theta_2) \tag{39}$$
$$v_i - v_{i-1} \sim \mathrm{Exp}(\theta_3) \tag{40}$$
$$d_i - d_{i-1} = s_i + \max(0, v_i - d_{i-1}). \tag{41}$$

In the above equations, $\mathcal{U}(\theta_1, \theta_2)$ denotes a uniform distribution in the range $[\theta_1, \theta_2]$, $\mathrm{Exp}(\theta_3)$ denotes an exponential distribution with rate $\theta_3$, and $v_0 = d_0 = 0$. In our experiments we used a total of $I = 50$ jobs.

The goal is to infer parameters $\boldsymbol{\theta} = (\theta_1, \theta_2, \theta_3)$ if the only knowledge is a set of percentiles of the empirical distribution of the interdeparture times $d_i - d_{i-1}$ for $i = 1, \ldots, I$. In our experiments we used 5 equally spaced percentiles. That is, given a set of $I$ interdeparture times $d_i - d_{i-1}$, we took $\mathbf{x}$ to be the 0th, 25th, 50th, 75th and 100th percentiles of the set of interdeparture times. Note that the 0th and 100th percentiles correspond to the minimum and maximum element in the set.

Since different percentiles can have different scales and strong correlations between them, we whitened the data on the basis of a pilot run. That is, we performed 100K pilot simulations, and recorded the mean vector and covariance matrix of the resulting percentiles. For each subsequent simulation, we calculated $\mathbf{x}$ from resulting percentiles by subtracting the mean vector and decorrelating and normalizing with the covariance matrix.

To generate observed data $\mathbf{x}_o$, we set the parameters to the following values

$$\theta_1 = 1, \quad \theta_2 = 5, \quad \theta_3 = 0.2 \tag{42}$$

and simulated the model to get $\mathbf{x}_o$. We evaluated inference algorithms by how well the true parameter values were retrieved, as measured by log probability under computed posteriors. Finally, the prior probability of the parameters was taken to be

$$\theta_1 \sim \mathcal{U}(0, 10) \tag{43}$$
$$\theta_2 - \theta_1 \sim \mathcal{U}(0, 10) \tag{44}$$
$$\theta_3 \sim \mathcal{U}(0, {}^1/_3), \tag{45}$$

which is uniform, albeit not axis-aligned, and of course includes the true parameters.