[Reviews · NeurIPS 2016]

Reviewer 1

Summary

The paper proposes to replace standard Monte Carlo methods for ABC with a method based on Bayesian density estimation. Although density estimation has been used before within the context of ABC, this approach allows for the direct replacement of sample-based approximations of the posterior with an analytic approximation. The novelty is similar in scope that of ABC with variational inference [1], but the approach discussed here is quite different. [1] Tran, Nott and Kohn, 2015, "Variational Bayes with Intractable Likelihood"

Qualitative Assessment

The paper is actually very well written and easy to understand. The level of technical writing is sufficient for the expert, while eliding unnecessary details. While paper heavily builds upon previous work, the key idea of proposition 1 is used elegantly throughout, both to choose the proposal prior and to estimate the posterior approximation. In addition, there are other moderate novel but useful contributions sprinkled throughout the paper, such as the extension of MDN to SVI. However, the authors must also discuss other work in the use of SVI with ABC, such as for example [1]. The paper lacks a firm theoretical underpinning, apart from the asymptotic motivation that Proposition 1 provides to the proposed algorithm. However, I believe that this is more than sufficient for this type of paper, and I do not count that as a negative, especially given the NIPS format [I doubt that an explanation of the model, experiments as well heavy theory could fit in the eight pages provided]. The experimental results are a good mix of simple examples and larger datasets, and are clearly presented. I also like how the authors disentangle the effect of selecting the proposal distribution from the posterior estimation. The plots are trying to take the taking effective sample size into account, but I am not sure that this is the best metric. After all, samples are purely computational beasts in this setting. Wouldn't it make more sense to measure actual CPU time?

Confidence in this Review

3-Expert (read the paper in detail, know the area, quite certain of my opinion)


Reviewer 2

Summary

In this paper the authors present an alternative approach to Approximate Bayesian Computation for (of course) models with intractable likelihood but from which mock datasets may be readily simulated under arbitrary parameters (within the prior support, etc etc.). The approach presented makes use of a flexible parametric modelling tool—the Mixture Density Network—to approximate the Bayesian conditional density on the parameter space with respect to the (mock) data; in this way the authors bring about a potentially powerful synthesis of ideas from the machine learning and statistical theory.

Qualitative Assessment

I believe this may be an outstanding paper as the approach suggested is well motivated and clearly explained; and my impression from the numerous worked examples is that it will very likely have an impact on the application of likelihood free methods, especially (but not exclusively) for problems in which the mock data simulations are costly such that efficiency of the sampler or posterior approximation scheme is at a premium (e.g. weather simulations, cosmological simulations, individual simulation models for epidemiology). It is worth noting here the parallel development within the statistics community of random forest methods for epsilon-free ABC inference targeting models for the conditional density (Marin et al., 1605.05537), which highlight the enthusiasm for innovations in this direction. I have a concern with the authors’ proof of Proposition 1 in that the term ‘sufficiently flexible’ is not explicitly described but should be, in which case sufficient conditions on the posterior for use of the MVN model could be easily identified. Naturally these will be rather restrictive so interest turns to understanding and identifying circumstances where the the approximation may be considered adequate or otherwise, and empirical metrics by which the user might be guided in this decision. Minor notes: - the comparison to existing work in Section 4 is well done (e.g. identification of regression-adjustment as a development in a similar direction); perhaps though it is worth noting that the ‘earliest ABC work’ of Diggle & Gratton (1984) was to develop a kernel-based estimator of the likelihood - in the introduction it is mentioned that “it is not obvious how to perform some other computations using samples, such as combining posteriors from two separate analyses”; a number of recent studies in scaleable Bayesian methods have been directed towards this problem (e.g. Zheng, Kim, Ho & Xing 2014, Scott et al. 2013, Minsker et al. 2014)

Confidence in this Review

3-Expert (read the paper in detail, know the area, quite certain of my opinion)


Reviewer 3

Summary

The authors propose to approximate the posterior of intractable models using a density estimator based on neural network. The main advantage, relative to ABC methods, is that it is not necessary to choose a tolerance. The innovative part is that they model the posterior directly, while a more common approach is to approximate/estimate the intractable likelihood. Hence, Proposition 1 is the main result of the paper, in my opinion. Starting from Proposition 1, several conditional density estimators could be used, and the authors use a Mixture Density Network. They then describe how the proposal prior and the posterior density are estimated, using respectively Algorithm 1 and 2. They illustrate the method with several simple examples, two of which have intractable likelihoods.

Qualitative Assessment

The most original part of the paper is Proposition 1, which is quite interesting. However, I have some doubts regarding the assumptions leading to formula (2). As explained in the appendix, this formula holds if q_theta is complex enough to make so that the KL distance is zero. Now, in a realistic example and with finite sample size, q_theta can't be very complex, otherwise it would over-fit. Hence, (2) holds only approximately. The examples are a bit disappointing. In particular, tolerance-based ABC methods suffer in high dimensions, hence I would have expected to see at least one relatively high dimensional example (say 20d). It is not clear to me that the MDN estimator would scale well as the number of model parameters or of summary statistics increases. The practical utility of the method depends quite a lot on how it scales, and at the moment this is not evident. My understanding is that the complexity of the MDN fit depends on the hyper-parameter lambda and on the number of components. The number of components was chosen manually, but the value of lambda is never reported. How was this chosen? I have some further comments. Section by section: - Sec 2.3 1. Is a proper prior required? 2. In Algorithm 1, how is convergence assessed? Because the algorithm seems to be stochastic. - Sec 2.4 1. The authors say: "If we take p̃(θ) to be the actual prior, then q φ (θ | x) will learn the posterior for all x" is this really true? Depending on the prior, the model might learn the posterior for values of x very different from x_0, but probably not "for all x". Maybe it is also worth pointing out that you need to model qφ(θ | x) close to x_0 because you are modelling the posterior non-parametrically. If, for instance, you were using a linear regression model, the variance of the estimator would be much reduced by choosing points x very far from x_0. 2. Why the authors use one Gaussian component for the proposal prior and several for the posterior? Is sampling from a MDN with multiple components expensive? If the same number of components was used, it might be possible to unify Algorithms 1 and 2. That is, repeat algorithm 2 several times, use the estimated posterior at the i-th iteration as the proposal prior for the next one. 3. It is not clear to me how MDN is initialized at each iteration in Algorithm 1. The authors say that by initializing the prior using the previous iteration allows them to keep N small. Hence, I think that by initializing they don't simply mean giving a good initialization to the optimizer, but something related to recycling all the simulations obtained so far. Either way, at the moment is it not quite clear what happens. - Sec 2.5 1. It is not clear to me why MVN-SVI avoids overfitting. Whether it overfits or not probably depends on the hyperparameter \lambda. How is this chosen at the moment? I guess not by cross-validation, given that the authors say that no validation set is needed. - Sec 3.1 1. The differences between the densities in the left plot of Figure 1 are barely visible. Maybe plot log-densities? 2. What value of lambda was used to obtain these results? This is not reported, same in the remaining examples. - Sec 3.2 1. Is formula (5) correct? x is a vector, but its mean and variance are scalar. 2. In Figure 2: maybe it is worth explaining why in ABC the largest number of simulations does not correspond to the smallest KL distance. I guess that this is because \epsilon is too small and the rejection rate is high. - Sec 3.3 1. The authors say that "in all cases the MDNs chose to use only one component and switch the rest off, which is consistent with our observation about the near-Gaussianity of the posterior". Does this happen for any value of \lambda?

Confidence in this Review

2-Confident (read it all; understood it all reasonably well)


Reviewer 4

Summary

This paper proposes a method for parameters inference. The paper sets the problem where we have a set of observed variables, x, and a set of underlying parameters theta. We assume that we can sample from p(x|theta) but that we don't have an explicit form for it. The goal is to recover the parameter posterior p(theta|x). We assume we have a prior distribution p(theta) over the parameters theta. The paper explains that must of the usual methods to solve this kind of problems is to replace p(x=x0|theta) by p(||x-x0|| < epsilon|theta) and use a sampling method, such as MCMC. However, they explain that it only approximates the true distribution when epsilon goes to 0, but at the same time the computing complexity grows to infinity. The proposed method is to directly train a neural network to learn p(theta|x) (renormalized by a known ratio of pt(theta) over p(theta), explained later). The network produces the parameters for a mixture of Gaussian. The training points are drawn from the following procedure: choose a distribution pt(theta) to sample from. Sample a batch a N points from pt(theta). Run them through the sampler to get the corresponding points x. Train the network to predict p(x|theta) from the input theta. The selection of pt is important for convergence speed, and a method is proposed: start with the prior p(theta) and as the neural network is trained, use the current model to refine the prior pt. Results are on multiple datasets, and the method seems to work well, and converge better than MCMC and simple rejection methods.

Qualitative Assessment

The paper is clear and the method looks sounds. Several related works are presented towards the end of the paper (why not the beginning as in most papers?). The differences between the current method and these are explained, but no comparisons are directly shown with most of the related methods. It would be nice to include these on at least one problem.

Confidence in this Review

1-Less confident (might not have understood significant parts)


Reviewer 5

Summary

The paper is on likelihood-free inference, that is on parametric inference for models where the likelihood function is too expensive to evaluate. It is proposed to obtain an approximation of the posterior distribution of the parameters by approximating the conditional distribution of data given parameters with a Gaussian mixture model (a mixture density network). The authors see the main advantages over standard approximate Bayesian computation (ABC) in that - their approach is returning a "parametric approximation to the exact posterior" as opposed to returning samples from an approximate posterior (line 49), - their approach is computationally more efficient. (line 55) The paper contains a short theoretical part where the approach is shown to yield the correct posterior in the limit of infinitely many simulations if the mixture model can represent any density. The approach is verified on two toy models where the true posterior is known and two more demanding models with intractable likelihoods.

Qualitative Assessment

Technical quality --------------------- My ranking is due to the following: 1. The authors say that their approach approximates or targets the exact posterior (e.g. lines 50; 291). I strongly disagree with the statement. The paper only concerns the approximation of the posterior after conversion of data to summary statistics. But the summary statistics are most often not sufficient so that important information is lost and targeting the "exact posterior" is not possible any more. 2. It is emphasized that the approximation can be made as accurate as required (line 51). I suppose this becomes possible when the number of components and hidden layer in the network model is increased. Unfortunately, model choice does not seem to get discussed: How should the user choose the number of layers and components, taking in consideration that more data need to be simulated to fit more complex models? 3. While I appreciate theoretical justifications, Proposition 1 did not provide me with much information. Asymptotically, rejection ABC recovers the posterior too, under even weaker conditions, because the threshold (bandwidth) decreases to zero as the number of simulations increases (e.g. Section 3.1. in Blum2010). In that sense, also in rejection ABC, the approximation "can be made as accurate as required". 4. Regression adjustment is a classical method in ABC that deals with the increase of computational cost when the bandwidth epsilon is decreased (see e.g. Section 4.1 of the 2010 review paper by Beaumont). Some of the work is briefly mentioned toward the end of the paper, but no experimental comparison is performed. Such a comparison, however, would have been very important, at least because - they are standard practice in ABC, - they aim at producing (approximate) posteriors for epsilon \to zero, like the proposed method, - there are straightforward theoretical connections to the proposed method (see below the comments on novelty). Further comments: 1. In Fig1 (left), the more advanced algorithm 2 (MDN with proposal) seems to produce a posterior that is less accurate than the simpler algorithm 1 (MDN with prior). Would you have an explanation for that? Is it not worrisome that the more expensive algorithm 2, which builds on algorithm 1, is able to decrease the accuracy of the approximation? 2. In sections 3.3 and 3.4., the log probability (density?) that the learned posterior assigns to the true parameter value is used to assess the accuracy of the method. I think this measure could be possibly misleading because in many cases, in particular for weakly informative data, the true posterior is not centered at the data generating parameter (consider e.g. the posterior of the mean of a Gaussian). Furthermore, it does not measure the accuracy of the spread of the posterior. Getting the spread (e.g. posterior variance) right, however, is important, as pointed by the authors (line 22). 3. It is rather common to work with uniform priors on a bounded interval (e.g. [0,1]). Is it correct that the estimated posterior pHat in (3) would then be a truncated mixture of Gaussians? For truncated (mixture of) Gaussians - exact normalization is not possible and numerically challenging in higher dimensions, - the marginals are generally not truncated (mixtures of) Gaussians (see e.g. Horrace2005) Can this be a problem for the proposed approach? 4. Eq (7) in the supplementary material shows that q_phi(theta|x_0)/pTilde(theta) equals the likelihood function (as the proportionality factor does not matter). Since pTilde is known and fixed in each iteration, does learning q_phi thus correspond to learning the likelihood function? Novelty/originality --------------------- I think the novelty claims of the paper are too strong. Related work is mentioned in section 4 but the discussion stays on the surface and the proposed method is not placed well in the existing body of research. It is presented as a "new approach to likelihood-free inference based on Bayesian conditional density estimation". But it seems to conceptually belong to existing likelihood-free inference approaches. 1. ABC has been approached via conditional density estimation since more than 10 years (see e.g. section 4.1 of the review by Beaumont 2010). This connection resulted in the methods for regression adjustment. All these methods target the approximate posterior in the limit of epsilon \to zero. 2. When linear models are used, a pilot ABC run is needed to restrict the parameter space, which results in discarding simulations. For nonlinear models, however, all simulations can be used (see e.g. Fig 1 of Blum's 2010 paper in Statistics and Computing, reference 5 in your paper). This also corresponds to "\epsilon-free inference". Moreover, Blum used neural networks to model the relation between data (summary statistics) and parameters as well. 3. The 2012 paper by Fan, Nott, and Sisson that is cited as reference 8 is an epsilon-free inference approach based on conditional density estimation too, and it also makes use of mixture models. While Fan et al focus on approximating the likelihood, the work by Bonassi et al (2011), discussed by Fan et al, focuses on approximating the posterior. 4. Likelihood-free inference methods can be classified into parametric and nonparametric approximations (see e.g. section 3 of reference 11). Classical ABC algorithms correspond to nonparametric approximations where the "epsilon" plays the role of the bandwidth. Parametric approximations, by construction, do not require such a bandwidth (an example is reference 12). The proposed method is a parametric approach so that it is not surprising that it is "epsilon-free". I agree that other researchers may not have used the particular neural network used in this paper. But it is unclear to me why the paper is presented as a "new approach to likelihood-free inference". It seems more like a technical difference to previous work, and its advantages relative to recent efficient likelihood-free inference methods are neither presented nor discussed. References ------------ Blum2010: Approximate Bayesian Computation: A Nonparametric Perspective, Journal of the American Statistical Association, 2010, 105, 1178-1187 Beaumont2010: Approximate Bayesian Computation in Evolution and Ecology, Annual Review of Ecology, Evolution, and Systematics, 2010, 41, 379-406 Horrace2005: W. Some results on the multivariate truncated normal distribution, Journal of Multivariate Analysis, 2005, 94, 209-221 Bonassi2011: Bayesian Learning from Marginal Data in Bionetwork Models, Statistical Applications in Genetics and Molecular Biology, 2011, 10. Update: ---------- Based on the explanation and promises in the author reply, I increased my scores from 2 to 3. I suggest to take the following points to heart when revising the paper. - Please clarify that the "exact posterior" refers to the distribution of the parameters given the summary statistics and not the actually exact posterior. The expression "exact posterior" is misleading. - Please clarify the difference to existing methods for likelihood-free inference that are based on (parametric) conditional density estimation, and update novelty claims accordingly. - Please acknowledge that previous work on likelihood-free inference was able to greatly reduce the computational cost too (e.g. ref 11, 15). - Please clarify why the approach is approximating the posterior rather than the likelihood function. In my opinion, Eq (3) shows that q_phi(theta|x_0)/ptilde(theta) must be proportional to the likelihood function. And thus by learning phi, you obtain an approximation of the likelihood function.

Confidence in this Review

3-Expert (read the paper in detail, know the area, quite certain of my opinion)